# Analysis and Minimization of Race Tracking in the Resin-Transfer-Molding Process by Monte Carlo Simulation

**DOI:** 10.3390/ma16124438

**Published:** 2023-06-16

**Authors:** Romain Agogué, Modesar Shakoor, Pierre Beauchêne, Chung Hae Park

**Affiliations:** 1IPC—Centre Technique Industriel de la Plasturgie et des Composites, Rue Léonard de Vinci, F-53810 Laval, France; 2IMT Nord Europe, Institut Mines-Télécom, Univ. Lille, Centre for Materials and Processes, F-59000 Lille, France; 3ONERA (The French Aerospace Lab), 29 Avenue de la Division Leclerc, F-92320 Chatillon, France

**Keywords:** race tracking, Monte Carlo simulation, dry spots, resin transfer molding (RTM), permeability

## Abstract

A numerical analysis of the influence of race tracking on dry spots formation and the accuracy of permeability measurement during the resin-transfer-molding process is presented. In the numerical simulation of the mold-filling process, defects are randomly generated, and their effect is assessed by a Monte Carlo simulation method. The effect of race tracking on the unsaturated permeability measurement and dry spots formation is investigated on flat plates. It is observed that the race-tracking defects located near the injection gate increase up to 40% of the value of the measured unsaturated permeability. The race-tracking defects located near the air vents are more likely to generate dry spots, whereas those near the injection gates have a less significant influence on dry spots generation. Depending on vent location, it has for instance been shown that the dry spot area can increase by a factor of 30. Dry spots may be mitigated by placing an air vent at a suitable location based on the numerical analysis results. Moreover, those results may be helpful to determine optimal sensor locations for the on-line control of mold-filling processes. Finally, the approach is successfully applied to a complex geometry.

## 1. Introduction

Liquid-composite-molding processes, such as resin-transfer molding (RTM) and vacuum-assisted resin-transfer molding (VARTM), have gained great interest during the past decades in different industrial sectors as cost-effective composites-manufacturing processes. For example, high-pressure resin-transfer molding (HP-RTM) and compression RTM (C-RTM) are considered in the automotive sector to fabricate light structural parts with high performance at a high production rate and a limited generation of volatile emissions. Despite those competitive advantages, these processes still suffer from a lack of robustness [1,2]. Some authors have investigated the origin of this randomness and have provided evidence that it may come from the variation of preform permeability [3] or the presence of empty flow channels without fibers either between the mold and the preform [4] or around internal edges, such as sharp corners or angles where the preform fails to conform to the exact mold shape [5,6]. Through these flow channels, the resin flow can advance much faster than through the fiber preform. This phenomenon is called the race-tracking effect and eventually makes the flow front progress highly irregular, which may lead to the generation of dry spots and an incorrect measure of the permeability [2]. Consequently, the injection and mold pressures may be lower than expected as well as the mold-filling time [7,8]. Although these negative effects could deteriorate the performance of the process and increase the scrap ratio, the race-tracking effect can also be exploited positively if it is controlled. Race-tracking channels can be used to control and improve the flow distribution and minimize the process cycle times [7].

The presence of these race-tracking channels can be mainly attributed to the error induced during the manufacturing process. Indeed, a typical composition of LCM manufacturing steps is (i.) preparation of a preform; (ii.) injection of liquid resin into the mold for the impregnation of the fiber preform; (iii.) curing of the resin; (iv.) release of the part from the mold. In most industrial implementations of RTM, C-RTM, and HP-RTM processes, a few key steps are still manually performed. For instance, fiber reinforcements are often cut and draped manually. This manual operation may lead to undesired preforming defects, such as empty spaces or open channels between the mold wall and the fiber preform. As a result, during the impregnation stage, race tracking is likely to take place, leading to dry spots in the final manufactured part [9]. Although it might be possible to reduce these defects by improving tolerances of the preform cutting and layup and relying on the experience of the operators, it is not possible in general to know in advance whether flow channels will be created in the mold or not and how they will be distributed. As a matter of fact, it is very difficult or almost impossible to predict the location, the size, and the number of race-tracking channels before the real manufacturing.

Therefore, some authors have proposed active control strategies to reduce the effect of race tracking on the overall mold-filling process [7,10,11]. For instance, if it is possible to track the resin flow front in real time at some locations in the mold, then an adaptive control algorithm can use this information to modulate the inlet pressure and drive the resin flow front to advance regularly as if no race tracking was present [12]. For molds with a complex shape, more advanced control strategies have been proposed where the resin flow front is tracked using a number of sensors placed at different locations in the mold, and the inlet pressure as well as the pressure at auxiliary gates and vents are modulated to control the flow and prevent dry spots [13]. Numerical simulation tools are playing an increasing role in recent control strategies because they can be used to predict the effect of race tracking for different scenarios without tedious mold-filling experiments in order to minimize the number of sensors and track the resin flow front more accurately [14,15,16]. Nevertheless, simulation tools can give information only for given conditions, such as the presumed location, size, and length of race-tracking channels. As a matter of fact, there may be an excessively great number of configurations even for a very simple mold geometry. It is practically impossible to manually select representative ones for the numerical simulation. Therefore, a systematic approach would be interesting to consider a large number of cases of race-tracking channels and improve the reliability of mold design and sensor optimization by numerical mold-filling simulations. 

The objective of this paper is to analyze the influence of the position and size of race-tracking channels on the overall mold-filling pattern and dry spots formation. This analysis is conducted for different mold configurations in order to provide guidance to process engineers for the placement of sensors and gates/vents. Thanks to a Monte Carlo approach, it is possible to conduct this analysis in terms of standard deviations as well as mean values to assess the variability of the effect of race tracking. This analysis is proposed first for rectangular molds to assess the influence of race tracking on the accuracy of permeability measurement and the mold-filling pattern. Then, the same approach is applied to a mold with a complex shape, which is representative of complex composite structures.

## 2. Methods

### 2.1. Mold Geometry

Three rectangular mold configurations are considered with different lengths and widths, i.e., 0.5 × 0.46 m^2^, 0.6 × 0.3 m^2^, and 0.7 × 0.15 m^2^. The aspect ratio, i.e., the length over the width, of each plate is 1.1, 2.0, and 4.7, respectively. A thickness of 2 mm is assigned for each geometry.

For each case, a constant pressure of 2 × $10^{5}$ Pa is applied to a line injection gate located along the left edge of the mold and a vacuum of 0 Pa to a vent located at the right edge of the mold, as shown in Figure 1. Two types of vents are considered. (i.) A line vent is placed all along the edge opposite to the injection gate, hereafter referred to as “line”, and (ii.) a point vent is located in the middle of the edge opposite to the injection gate and can model as a single point, hereafter referred to as “middle.” The walls of the mold are modeled by a no-penetration condition, meaning there is no fluid passing through solid boundaries.

An overview of the simulated configurations is presented in Table 1. These configurations are similar to the one used in Ref. [12], both in terms of injection pressure and mold dimensions. Although a wider range of pressure gradients and mold shapes and dimensions are encountered in real manufacturing processes, these configurations are chosen first for validation of the proposed method. An application to a more complex and realistic mold is presented in Section 3.4.

### 2.2. Material Properties

A fiber preform with an isotropic permeability of $10^{-10}$ $m^{2}$ and a fiber volume fraction of 0.4 is considered. The gaps due to the race-tracking channels are modeled as a highly permeable material. Many authors have proposed models for the equivalent permeability of gaps [4,17]. In this paper, the channel is assumed to have a rectangular cross section, and the equation presented in [4] is adopted:(1)$$K=\frac{a^{2}}{12}\left(1-\frac{192a}{\pi^{5}b}\sum_{i=1,3,5\ldots }^{\infty }\frac{tanh\left(i\pi b/2a\right)}{i^{5}}\right)$$
where a and b are respectively the width and the height of the rectangular gap. The width is set to 5 mm, and the height is assumed to be the same as the height of the cavity. A permeability of 2.5 × $10^{-7}$ $m^{2}$ has been estimated, using Equation (1). Hammani et al. [17] obtained a similar value for the modeling of the race-tracking effect. They found good agreement in terms of flow front shape and position between the numerical predictions and the experimental observations.

To model the race tracking, the gaps are randomly placed around one edge of the mold geometry. The process for generating a race-tracking channel consists of three main steps:Step 1: All the elements of the mesh that could be included in the race-tracking channel are sorted and registered in a list.Step 2: Two random numbers are generated. The first one, generated with a uniform distribution, defines the center of the race-tracking channel. The second one, generated with a lognormal distribution (Equation (2)), defines the length of the race-tracking channel. While there have been discussions in the literature on the proper probability distribution for permeability [3,8,18,19], there are no studies regarding race-tracking channels. The distributions chosen herein for the channel’s center and length could easily be changed based on experimental measurements.Step 3: For each element registered in the list in step 1, if the distance between the element and the center of the race-tracking channel is lower than half the length, then the properties (e.g., equivalent permeability) of the race-tracking channel are assigned to the element. Otherwise, the properties of the preform are assigned.

This process is repeated for each Monte Carlo computation. The parameters used for the lognormal distribution are μ = −3.0 and σ = 0.5. With the chosen parameters, the average length where the race-tracking zone is included is 0.05 m.
(2)$$r\left(x\right)=\frac{1}{x\sigma \sqrt{2\pi }}e^{-\frac{{\left(\ln x-\mu \right)}^{2}}{2\sigma^{2}}}$$

The viscosity of the resin is set to 0.1 Pa·s, as in Refs. [3,19]. Note again that the distribution parameters should be reconsidered based on experimental data in future work, and the influence of the viscosity could also be analyzed.

### 2.3. Monte Carlo Simulation Method

One of the first statistical characterization studies of composites permeability has been conducted using an automated testbed [8]. Since then, numerical methods have gained increasing popularity as they enable testing many configurations at a reduced cost [19]. This is particularly relevant for the Monte Carlo method, which typically requires a large number of simulations to obtain relevant results [18]. Despite its computational cost, this method is chosen in the present work as it can provide an accurate statistical characterization [18].

For each case, one thousand race-tracking zones are randomly generated using Equation (2). Mold-filling-process simulations are conducted using the CV-FEM (control volume finite element method) with the FINE (floating imaginary nodes and elements) method [20]. As the Monte Carlo computations require huge computational resources, a grading technique is applied for the mesh refinement, as shown in Figure 1. The mesh density is higher near the race-tracking zone and coarser elsewhere. The number of elements for the 0.5 × 0.46 m^2^ is 3920, 4690 for the 0.6 × 0.3 m^2^, and 3455 for the 0.7 × 0.15 m^2^. 

### 2.4. Post-Processing

The following global indicators are used to evaluate the effect of gaps on the impregnation stage for each calculation:Overall filling fraction: The mold-filling process is considered terminated as soon as the resin has reached at least one of the vent nodes. The overall filling fraction is then estimated by computing the ratio of the mold volume filled by the resin to the total mold volume at this stage of the process.Unsaturated permeability (K): The unsaturated permeability is computed by monitoring the average flow position (in X axis) in time using the following equation for rectilinear flow:
(3)$$K=\frac{\mu \left(1-v_{f}\right)}{2P_{inj}t}{\bar{x}}^{2}\left(t\right)$$
where $\bar{x}\left(t\right)$ is the average flow position along the width of the mold (i.e., averaged along the Y axis) [m], $P_{inj}$ is the injection pressure [Pa], μ is the dynamic viscosity of the resin [Pa.s], and $v_{f}$ is the fiber volume fraction. Following the work of Gokce et al. [4] and Siddig et al. [15], the effective permeability, K, obtained by the aforementioned Equation (3) is then divided by the real permeability of the preform $K_{ref}=10^{-10}\;m^{2}$ to obtain the ratio, $K/K_{ref}$, which is monitored. The flow front position is computed from the fill factor, $s\left(x,y,z,t\right)$, by using the following equation, assuming V and w are respectively the volume and the width of the mold:(4)$$\bar{x}\left(t\right)=\frac{1}{w}\int_{V}s\left(x,y,z,t\right)dv$$Statistical fluctuation of flow front position and pressure field: the first two statistical moments (i.e., mean and standard deviation) are employed as indicators of statistical fluctuation. The outputs are the maps of mean and standard deviation of the flow front position and the pressure field over 1000 calculations, which are similar to the ones obtained by Zhang et al. [3].

## 3. Results and Discussion

### 3.1. Convergence Rate of the Monte Carlo Simulation Method

First, the convergence rate of the Monte Carlo approach is assessed. The mold with 0.6 × 0.3 m^2^ and a point vent placed in the center of the right edge is selected. Two probes, as virtual pressure sensors, are placed in the middle of the race-tracking zone at the coordinates (0.3, 0.3) and in the middle of the perform at the coordinates (0.3, 0.15). The pressure at the end of the mold-filling process is registered with some dispersion due to the random race-tracking zones. The average value and the standard deviation are estimated by fitting the numerical data to a Gaussian curve. The results are presented in Table 2.

The reference is the result from 1000 Monte Carlo computations. The convergence rate for estimating the average value is relatively high. After 10 Monte Carlo computations, an error close to 1% has been obtained for both probes. The convergence rate for estimating the standard deviation is lower. At least 500 computations are required to obtain an estimate with an error lower than 5%. Therefore, as all the computations presented in this paper have been performed with 1000 computations, they can be considered as converged.

### 3.2. Effect of Race Tracking on the Permeability Measurement

The effect of the race tracking on the accuracy of permeability measurement has been assessed. The results are presented in Table 3. As we consider the unsaturated permeability based on the flow front progress, the shape of the vent (i.e., line or point) has no influence on the result. Then, only the test cases with the point vent located in the middle of the right edge are presented. On the left side of Table 3, the computed distribution of permeability ratio (*K*/*K_ref_*) is presented for the three rectangular molds whose aspect ratio (AR) is indicated. The shape of the distribution is similar for the three molds. Overall, a permeability ratio ranging from 1 to 1.4 has been obtained. Lawrence et al. [21] conducted both simulations and experiments with similar rectangular molds and boundary conditions and obtained a ratio varying from 1.6 to 2.5 in their experimental work. These values are consistent with those found in the current study. Our values are lower as we consider gaps covering only a portion of the edge, whereas Lawrence et al. placed a gap all along the edge in the mold-filling experiments, leading to higher ratios.

On the right side of Table 3, a plot of the influence of the location of the race-tracking channel along the edge and its length on the permeability measurement is displayed. For each mold aspect ratio, both the length and the position of the race-tracking channel have a clear influence on the unsaturated permeability value. As the mold aspect ratio increases, the region where the permeability ratio is larger than 1.05 grows, which shows that the permeability ratio increases and the unsaturated permeability is overestimated. As the length of the race-tracking zone increases, the permeability ratio also increases, regardless of the mold aspect ratio. A significant increase of the permeability is observed when both the mold aspect ratio and the length of the race-tracking channel are increased. The permeability ratio becomes close to one, however, when the race tracking zone is placed near the vent or far from the injection gate. Short race-tracking channels or ones located close to the vent have a limited influence on the permeability measurement. To obtain an accurate permeability value, therefore, special care should be taken to avoid race-tracking-channel formation in the upstream of the flow zone during the placement of a preform in the mold.

### 3.3. Effect of Race Tracking on Dry Spots Formation

The effect of race tracking on dry spots formation has also been investigated on the three rectangular molds with different aspect ratios and for the two types of vent (viz. line and point). The results are presented in Table 4. Each dot refers to a single calculation. The size and the color of each dot are representative of the surface of a dry spot, which is the ratio of the dry spot surface to the whole mold surface. For each result, the same color map has been used with a threshold of 4%. First, it has been observed that the shape of the vent (viz. line or point) highly influences the generation of dry spots. The point vent located in the middle of the edge exhibits a more robust behavior with race tracking than the line vent located all along the edge. Overall, dry spot area is reduced by a factor between 10 and 30 when using the point vent instead of the line vent. This is consistent with the observation by Li et al. [18] that placing the vent at the location where the flow ends prevents dry spots formation.

Race-tracking channels located close to the vent also increase the risk of generating dry spots compared to channels located near the injection gates, especially when the mold aspect ratio is high. For a mold aspect ratio of 4.7, race-tracking channels located at a distance below 0.2 m from the vent generate larger dry spots with a surface above 4% of the mold surface. This distance between the vent and the race-tracking channel, which leads to a high portion of dry spots (i.e., 4% assumed in this work), is 0.3 m for the mold aspect ratio of 2.0. No such minimal distance has been observed for the mold with the aspect ratio of 1.1.

Race-tracking channels located near the injection gate have a low chance of generating dry spots. The reason is that after flowing into the gap, the resin flows from the race-tracking channel to the preform as well as from the injection gates to the vent through the channel. Thus, this transverse flow from the race-tracking channel to the fiber preform helps the flow front in the preform to advance faster and catch up with the flow lead–lag generated by the race-tracking effect, as shown in Figure 2.

### 3.4. Application to a Complex Shape Mold

A complex stiffened panel is selected in the second part of this study (Figure 3). The outer dimensions of the panel are 0.63 × 0.6 m^2^, the height of the stiffener is 70 mm, and the thickness of this panel is 2 mm. The edges located at the base of the stiffener have a radius of 20 mm. The mold-filling-process parameters are the same as those adopted for the rectangular molds, as presented in Table 5.

For this complex mold geometry, only the internal race-tracking channels located at the base of the stiffener are considered. For the gaps located at corners, Dong [5] reported a maximal gap height ranging from 0.15 to 0.8 mm for a glass fiber bed with a fiber volume fraction of 0.4. Now, assuming a gap width of 10 mm and applying Equation (1), the permeability of corner-gaps ranges from $2\times 10^{-9}$ to $5\times 10^{-8}$ $m^{2}$. In this work, a permeability of $10^{-8}$ $m^{2}$ is assigned to the race-tracking zone.

The effect of race tracking on dry spots formation is presented in Figure 4. The orientation of the race-tracking channels strongly influences the dry spot size. It is observed that the race-tracking channels oriented parallel to the resin flow highly increase the risk of dry spots formation, whereas the race-tracking zones oriented perpendicular to the resin flow have negligible influence on dry spots formation. The dry spot area, indeed, is increased by a factor between two and ten when race-tracking channels are oriented parallel to the resin flow instead of perpendicular.

As opposed to what has been observed for the rectangular molds, there is no evidence that the location of race-tracking channel, viz. near the injection gate or the vent, influences dry spots generation for the stiffened panel. It is therefore recommended to avoid having edges wherever race tracking is likely to occur, aligned with the flow front propagation.

Finally, the standard deviation on the pressure field computed at the end of the process is presented in Figure 5. The zones in this map with high standard deviations are optimal locations for pressure sensors. We can see that the edges aligned in the parallel direction of the main resin flow exhibit greater standard deviations, which may lead to an anomaly of the mold-filling pattern and eventual dry spots formation. Hence, some sensors to monitor the resin pressure or resin flow arrival may be placed in these regions in the case of on-line control of the mold-filling process.

## 4. Conclusions

The effect of race tracking on the unsaturated permeability and dry spots formation was analyzed by Monte Carlo simulation. For rectangular molds, the influence of the length and position of race-tracking channels was investigated. It has been observed that race-tracking channels located near the injection gate led to an overestimation of the unsaturated permeability with an increase up to 40%. On the contrary, race-tracking channels located near the vent were more likely to generate dry spots. For rectangular molds, it has also been shown that relying on an adequately placed point vent instead of a line vent could reduce dry spot area by a factor of at least 10. The case of a stiffened panel was also investigated. For that geometry, it was demonstrated that race-tracking channels aligned along the main resin flow direction were susceptible to dry spots formation with an increase of the dry spot area between two and ten as compared to race-tracking channels perpendicular to the flow direction. Thus, dry spots formation might be mitigated by placing another vent at a suitable location or sensors for on-line control of the mold-filling process. The present numerical approach by Monte Carlo simulation can be applied to design the process determining optimal vent locations as well as optimal flow sensor locations in order to decrease the risk of dry spots formation, while avoiding expensive and time-consuming experimental work.

In practice, a number of layers of fiber reinforcement are stacked to obtain a preform in the RTM process. Indeed, the resin flow in a multi-layer reinforcement can be three-dimensional (3D) because the permeability of each layer can be different. Since the computational cost associated with 3D flow simulation is extremely great, the resin flow in a multi-layer reinforcement is often considered as a two-dimensional plug flow with an average permeability of preform in the literature [22]. As a new efficient numerical method for 3D flow simulation in a multi-layer reinforcement has recently been developed in the literature [23], the optimization of race tracking in a multi-layer reinforcement could be an interesting topic in the future.

Moreover, the heat transfer and the corresponding viscosity change may lead to a significant influence on the mold-filling process when a mold-heating method is employed. The minimization of race tracking in the RTM process by local heating has been dealt with by experimental investigation in the literature [24,25]. In the current work, isothermal mold-filling simulation was employed, assuming a constant viscosity. Meanwhile, the numerical modeling of RTM taking into account the couplings among the resin flow, the heat transfer, the resin curing, and the viscosity change has well been addressed in the literature [26,27]. Therefore, the current methodology can be applied to the minimization of race tracking in the RTM process by thermal management with a mold-filling simulation considering the heat transfer, the resin curing, and the viscosity change in the future work.

Recently, some thermoplastic monomers for in-situ polymerization have been developed as an effective way to address the recycling issue of thermoset resins [28]. During the mold-filling process, these thermoplastic monomers maintain a viscosity as low as their thermoset counterparts. Therefore, the mold-filling process itself is equivalent for thermoplastic monomers and thermoset resins. As a consequence, the current approach using the Monte Carlo method has great potential for the mold-filling process optimization of thermoplastic RTM process as well.

## Figures and Tables

**Figure 1 materials-16-04438-f001:**
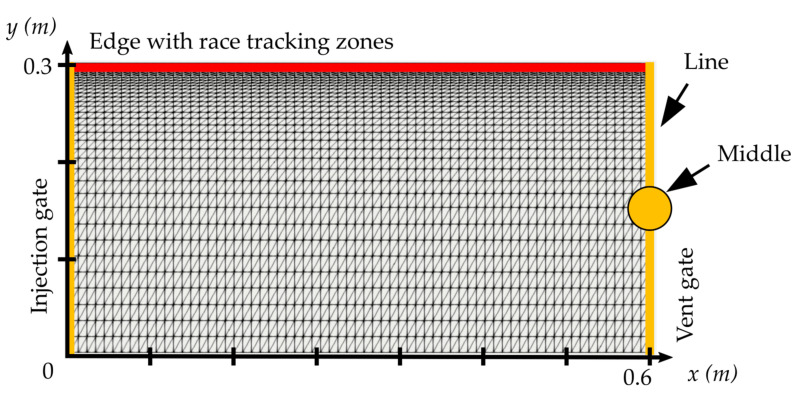
Details on the geometry, boundary conditions, and mesh grading—example for the geometry $0.6\times 0.3\;m^{2}$.

**Figure 2 materials-16-04438-f002:**
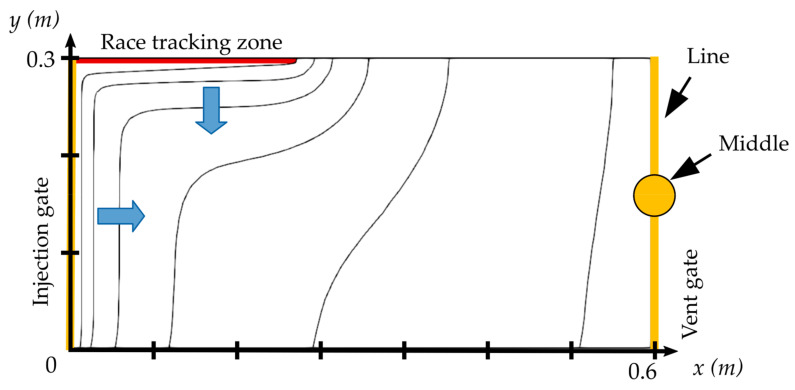
Front position at different times for a defect of length 18.6 cm for the $0.6\times 0.3\;m^{2}$ geometry.

**Figure 3 materials-16-04438-f003:**
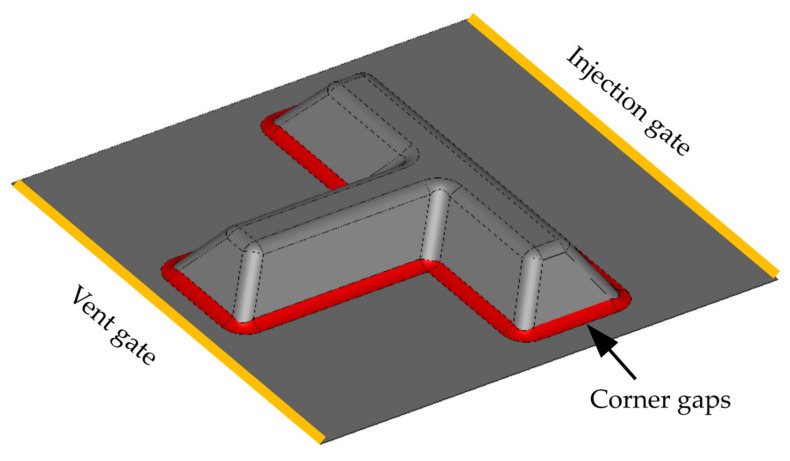
Geometry of the T-shape, including the position of the possible defects.

**Figure 4 materials-16-04438-f004:**
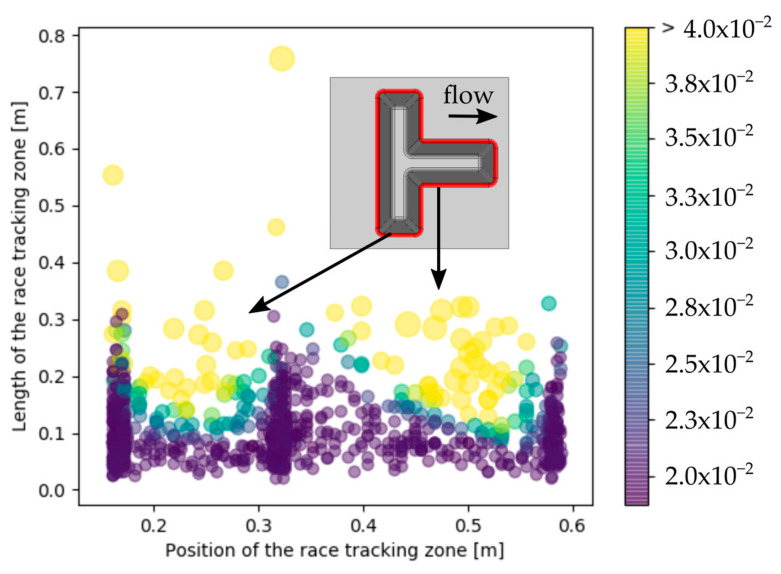
Effect of race tracking on the area of the dry spot (%) applied to the stiffened panel.

**Figure 5 materials-16-04438-f005:**
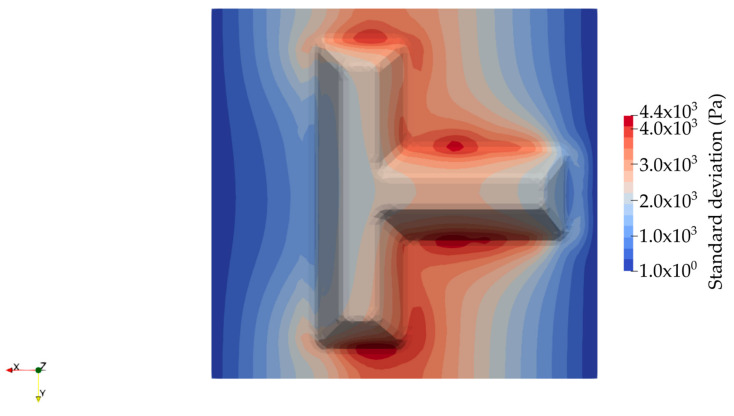
Standard deviation on the pressure field (Pa) computed at the end of the process.

**Table 1 materials-16-04438-t001:** Overview of the test cases.

Geometry	Dimensions	Aspect Ratio	Injection Parameter	Vent Geometry	Number of Computations
Plate	$0.5\times 0.46\;m^{2}$	1.1	$P=2\times 10^{5}$ Pa	Line	1000
Plate	$0.5\times 0.46\;m^{2}$	1.1	$P=2\times 10^{5}$ Pa	Middle	1000
Plate	$0.6\times 0.3\;m^{2}$	2.0	$P=2\times 10^{5}$ Pa	Line	1000
Plate	$0.6\times 0.3\;m^{2}$	2.0	$P=2\times 10^{5}$ Pa	Middle	1000
Plate	$0.7\times 0.15\;m^{2}$	4.7	$P=2\times 10^{5}$ Pa	Line	1000
Plate	$0.7\times 0.15\;m^{2}$	4.7	$P=2\times 10^{5}$ Pa	Middle	1000
Complex shape	$0.63\times 0.6\;m^{2}$		$P=2\times 10^{5}$ Pa	Line	1000

**Table 2 materials-16-04438-t002:** Mean and standard deviation of a probe located in the middle of the race-tracking zone, coordinates (0.3,0.3) for the geometry $0.6\times 0.3\;m^{2}$.

.Nb. of Computations	Pressure—Mean		Pressure—Standard Deviation	
	Value [Pa]	% of the Reference	Value [Pa]	% of the Reference
Probe located in the middle of the race tracking zone, coordinates (0.3, 0.3)				
10	9.83 × 10^4^	0.27%	5.74 × 10^3^	34.48%
100	9.73 × 10^4^	1.23%	8.47 × 10^3^	3.28%
400	9.71 × 10^4^	1.46%	7.97 × 10^3^	8.96%
500	9.73 × 10^4^	1.25%	8.44 × 10^3^	3.63%
700	9.86 × 10^4^	−0.05%	8.88 × 10^3^	−1.43%
1000	9.85 × 10^4^	Reference	8.76 × 10^3^	Reference
Probe located in the middle of the preform, coordinates (0.3, 0.15)				
10	9.39 × 10^4^	4.66%	8.76 × 10^3^	−0.23%
100	9.73 × 10^4^	1.24%	8.47 × 10^3^	3.07%
400	9.71 × 10^4^	1.47%	7.97 × 10^3^	8.77%
500	9.73 × 10^4^	1.26%	8.44 × 10^3^	3.42%
700	9.86 × 10^4^	−0.04%	8.88 × 10^3^	−1.64%
1000	9.85 × 10^4^	Reference	8.74 × 10^3^	Reference

**Table 3 materials-16-04438-t003:** Effect of race tracking on the unsaturated permeability.

Aspect Ratio	$\mathrm{Distribution}\;K/K_{ref}$	$\mathrm{Contour}\;\mathrm{Plot}\;K/K_{ref}$
1.1	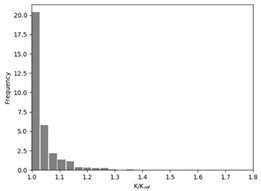	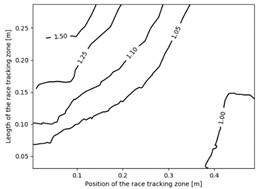
2.0	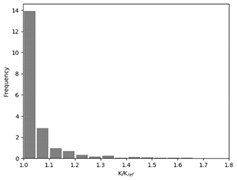	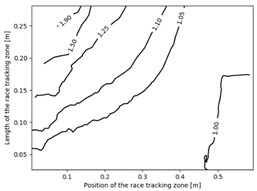
4.7	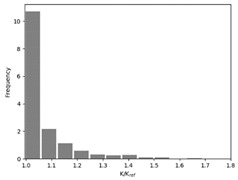	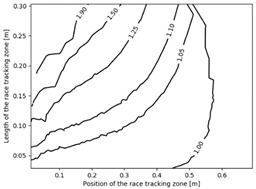

**Table 4 materials-16-04438-t004:** Effect of race tracking on the area of the dry spot (%).

Aspect Ratio	Vent Line	Vent Middle
1.1	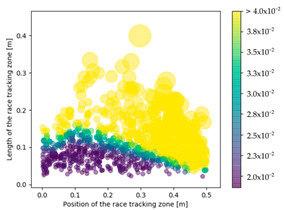	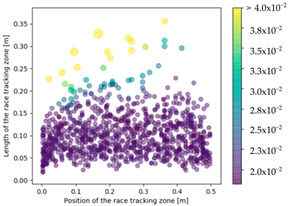
2.0	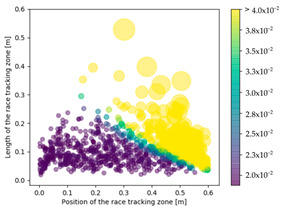	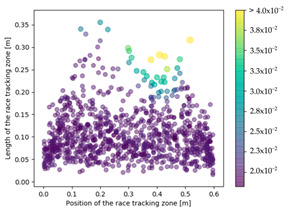
4.7	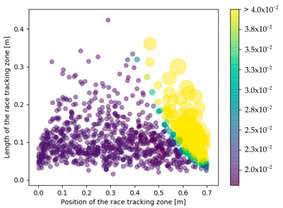	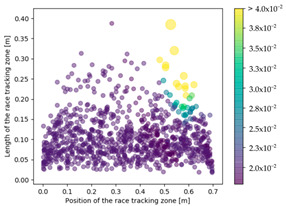

**Table 5 materials-16-04438-t005:** Numerical values for the CV-FEM computations.

Parameter	Unit	Value
Fiber volume fraction	-	0.4
Permeability of the perform	$m^{2}$	$10^{-10}$
Permeability of the edge-gaps	$m^{2}$	$2.5\times 10^{-7}$
Permeability of the corner-gaps	$m^{2}$	$10^{-8}$
Viscosity of the resin	Pa·s	0.1
Injection pressure	Pa	$2\times 10^{5}$
Vent pressure	Pa	0
Average length of defects	m	0.1

## Data Availability

The data presented in this study are available on reasonable request from the corresponding author.

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
