# Peer review of "Analysis and Minimization of Race Tracking in the Resin-Transfer-Molding Process by Monte Carlo Simulation"

_materials, 2023, doi:10.3390/ma16124438_

Round 1

Reviewer 1 Report

This paper stduy the race tracking in the resin transfer molding process by Monte-Carlo simulation, which has potential application value in engineering. In order to meet the requirements of high-quality publication of the "Materials" journal, it is recommended to consider the following suggestions.

1) There is no key quantitative data in the abstract.

2) What is the basis for selecting process parameter values in Table 1?

3) Line 108 of "An overview of the simulated configurations is presented in Table 1.". I think the paragraph format is not very good.

4) The mehtod proposed in this paper needs to be compared with the previous literature, otherwise it cannot reflect innovation.

5) The Discussion Section is missed.

6) The conclusion lacks quantitative data.

7) There are few references in the last three years.

8) The DPI of all Figures in the paper needs to be improved.

Author Response

This paper stduy the race tracking in the resin transfer molding process by Monte-Carlo simulation, which has potential application value in engineering. In order to meet the requirements of high-quality publication of the "Materials" journal, it is recommended to consider the following suggestions.

1) There is no key quantitative data in the abstract.

Quantitative data has been added to the abstract in the revised paper as suggested.

2) What is the basis for selecting process parameter values in Table 1?

These parameters have been chosen based on the literature to be representative of laboratory molds used by researchers. A comment has been added in page 3 of the revised paper to mention that.

3) Line 108 of "An overview of the simulated configurations is presented in Table 1.". I think the paragraph format is not very good.

It has been fixed in the revised paper.

4) The mehtod proposed in this paper needs to be compared with the previous literature, otherwise it cannot reflect innovation.

Numerical methods for RTM mold filling simulation have well been reported in the literature [Park, C.H.: 15 - Numerical simulation of flow processes in composites manufacturing. In: Boisse, P. (ed.) Advances in Composites Manufacturing and Process Design, pp. 317–378. Woodhead Publishing (2015). https://doi.org/10.1016/B978-1-78242-307-2.00015-4]. The control volume finite element method coupled with the volume of fluid method was adopted in this work, because it is considered the standard method for RTM mold filling simulation and it is commonly employed in most of commercial codes. It should be noted that the objective of the current method is not to develop a mold filling simulation method but to apply Monte-Carlo method for the analysis and optimization of race tracking problems. So far, the common approach to analyze the race tracking effects has been an experimental investigation. In this case, the number of test cases is relatively small and it is hard to perform exhaustive investigation. Up to our knowledge, it is the first time a great number of test cases are generated by numerical simulations by dint of Monte-Carlo method. Therefore, we have not found any previous work on the extensive investigation of race tracking effect by numerical methods. For the moment, no relevant reference has been found to compare with the current approach. Some comments on this point are made in the introduction of the revised paper.

5) The Discussion Section is missed.

We have chosen to combine the Results and Discussion sections, as authorized by Materials’ guidelines. To make this clearer, however, we have combined sections 3 and 4 in the revised paper.

6) The conclusion lacks quantitative data.

Quantitative data has been added to the conclusion in the revised paper as suggested.

7) There are few references in the last three years.

We have found no study similar to ours in the last years. There are, nevertheless, some studies such as reviews on resin transfer molding or papers on optimization and control strategies to reduce the influence of race-tracking that we have cited in pages 1-2 of the revised paper.

8) The DPI of all Figures in the paper needs to be improved.

We have improved the Figures as suggested.

Reviewer 2 Report

RTM has high interest; however, the use of thermosets strongly conditionate the future of this technique due to the high environmental impact. The authors should address some additional information regarding the processing of thermoplastic by RTM (T-RTM techniques) and how can this Monte-Carlo computation method be extended?

Regarding the presented “complex shape mold” I would classified it as a low complex shape mold. The size font in figures is very large (as I see in my computer)

Author Response

RTM has high interest; however, the use of thermosets strongly conditionate the future of this technique due to the high environmental impact. The authors should address some additional information regarding the processing of thermoplastic by RTM (T-RTM techniques) and how can this Monte-Carlo computation method be extended?

Of course, the current Monte-Carlo method can be applied to the thermoplastics’ RTM process as well. In general, thermoplastic polymer melts have much greater viscosities than thermoset counter parts. Recently, some thermoplastic monomers for in-situ polymerization whose viscosities are as low as those of thermoset resins have been developed for the RTM process [Bodaghi, Masoud, Chung Hae Park, and Patricia Krawczak. "Reactive processing of acrylic-based thermoplastic composites: A mini-review." Front. Mater. 9: 931338. doi: 10.3389/fmats (2022).]. Therefore, the mold filling process itself is almost equivalent for both thermoset and thermoplastic monomer resins. We added some comments on this subject in the conclusions of the revised paper.

Regarding the presented “complex shape mold” I would classified it as a low complex shape mold.

See our response to Reviewer #3’s 4th comment, which is copied hereafter: “More complex situations involving multi-layer reinforcements would definitely be worth considering. Indeed, the resin flow in a multi-layer reinforcement can be three-dimensional (3D) because the permeability of each layer can be different. Because the computational cost associated with 3D flow simulation is extremely great, the resin flow in a multi-layer reinforcement has been considered as a two-dimensional plug flow employing an average permeability of preform in an RTM process optimization in the literature [Park, C. H., Saouab, A., Bréard, J., Han, W. S., Vautrin, A., & Lee, W. I. (2009). An integrated optimisation for the weight, the structural performance and the cost of composite structures. Composites Science and Technology, 69(7-8), 1101-1107]. As a new efficient numerical method for 3D flow simulation in a multi-layer reinforcement has recently been developed in the literature [Chebil, N., Deleglise-Lagardere, M., & Park, C. H. (2019). Efficient numerical simulation method for three dimensional resin flow in laminated preform during liquid composite molding processes. Composites Part A: Applied Science and Manufacturing, 125, 105519], the optimization of race-tracking in a multi-layer reinforcement can be an interesting topic in the future work.  We believe, however, that the results presented in the current paper are sufficient to prove the validity of our method and its capabilities.”

The size font in figures is very large (as I see in my computer)

We have improved the Figures as suggested.

Reviewer 3 Report

The investigation of dry spots formation in polymer composite with fabric reinforcement is considered in the paper "Analysis and minimization of race tracking in the resin transfer molding process by Monte-Carlo simulation". The results of mathematical modeling by Monte-Carlo method are showing in that paper. It is showing that the dry spots arrears can be detected by using this mathematical modeling method. There is undoubtedly practice interest in the results of this paper. But there are some questions, some of them listed below:

1. The distributions types have to be justifying (lines 126-129). So, as the parameters of distributions should be justified too (lines 135-137).

2. Why the Monte Carlo method was chosen.

3. How does the viscosity of the resin affect to the results. The reviewer recommends the authors to conduct these studies.

4. The authors consider only a rectangular cell of the reinforcing fabric. What would be the results when a different cell shape was used? What about the results when multi-layer reinforcement is used.

Author Response

The investigation of dry spots formation in polymer composite with fabric reinforcement is considered in the paper "Analysis and minimization of race tracking in the resin transfer molding process by Monte-Carlo simulation". The results of mathematical modeling by Monte-Carlo method are showing in that paper. It is showing that the dry spots arrears can be detected by using this mathematical modeling method. There is undoubtedly practice interest in the results of this paper. But there are some questions, some of them listed below:

  1. The distributions types have to be justifying (lines 126-129). So, as the parameters of distributions should be justified too (lines 135-137).

We have added justifications based on the literature in page 4 of the revised paper as suggested.

  1. Why the Monte Carlo method was chosen.

We have explained this choice based on the literature in page 4 of the revised paper as suggested.

  1. How does the viscosity of the resin affect to the results. The reviewer recommends the authors to conduct these studies.

As explained in Ref. [16] of the revised paper, viscosity variations due, for instance, to temperature, have a significant influence on dry spot formation and can have a combined effect with race-tracking.  As the reviewer indicates, this could be interesting to study but we believe it can be left for future work. We have justified our choice of viscosity based on the literature in page 4 of the revised paper, as well as a comment on its potential influence. Some comments about the perspectives on this subject have been made in the section of conclusion, as well.

  1. The authors consider only a rectangular cell of the reinforcing fabric. What would be the results when a different cell shape was used? What about the results when multi-layer reinforcement is used.

There is already an analysis on a complex shape mold in the paper (Subsection 3.4 in the revised paper). This analysis shows that the orientation of race tracking channels with respect to the resin flow direction also plays a role. More complex situations involving multi-layer reinforcements would definitely be worth considering. Indeed, the resin flow in a multi-layer reinforcement can be three-dimensional (3D) because the permeability of each layer can be different. Because the computational cost associated with 3D flow simulation is extremely great, the resin flow in a multi-layer reinforcement has been considered as a two-dimensional plug flow employing an average permeability of preform in an RTM process optimization in the literature [Park, C. H., Saouab, A., Bréard, J., Han, W. S., Vautrin, A., & Lee, W. I. (2009). An integrated optimisation for the weight, the structural performance and the cost of composite structures. Composites Science and Technology, 69(7-8), 1101-1107]. As a new efficient numerical method for 3D flow simulation in a multi-layer reinforcement has recently been developed in the literature [Chebil, N., Deleglise-Lagardere, M., & Park, C. H. (2019). Efficient numerical simulation method for three dimensional resin flow in laminated preform during liquid composite molding processes. Composites Part A: Applied Science and Manufacturing, 125, 105519], the optimization of race-tracking in a multi-layer reinforcement can be an interesting topic in the future work.  We believe, however, that the results presented in the current paper are sufficient to prove the validity of our method and its capabilities.